# Microstructural Changes in the Macula Following Cataract Surgery in Patients with Type 2 Diabetes Mellitus Detected Using Optical Coherence Tomography Angiography

**DOI:** 10.3390/diagnostics13040605

**Published:** 2023-02-07

**Authors:** Lelde Svjaščenkova, Guna Laganovska, Lilian Tzivian

**Affiliations:** 1Department of Doctoral Studies, Riga Stradins University, LV1007 Riga, Latvia; 2Pauls Stradins Clinical University Hospital, LV1002 Riga, Latvia; 3Department of Ophthalmology, Riga Stradins University, LV1007 Riga, Latvia; 4Faculty of Medicine, University of Latvia, LV1586 Riga, Latvia; 5Holon Institute of Technology, Holon 5810201, Israel

**Keywords:** optical coherence tomography angiography, diabetic retinopathy, diabetic maculopathy, phacoemulsification cataract surgery, vessel density, foveal avascular zone

## Abstract

Background. Ophthalmologists treat diabetic macular edema before cataract surgery to reduce possible complications. Despite improvements in diagnostic techniques, whether cataract surgery per se causes the progression of diabetic retinopathy with macular edema remains unclear. This study aimed to evaluate the impact of phacoemulsification on the central retina and its correlation with diabetes compensation as well as changes in the retina before surgery. Methods. Thirty-four type 2 diabetes mellitus patients who underwent phacoemulsification cataract surgery were included in this prospective longitudinal study. Of them, 29.4% had macular edema before surgery, and 70.6% had a normal macular structure. All patients underwent ophthalmic examinations, including optical coherence tomography angiography, at baseline and at one and three months after surgery. The Mann-Whitney test was performed to compare the area of the foveal avascular zone, perimeter of the foveal avascular zone, and mean vascular density in the para- and perifoveal deep and superficial capillary plexuses. All parameters were measured before and at one and three months after surgery. Multiple linear regression models with adjustments for glycated hemoglobin and duration of diabetes mellitus were constructed to assess the association between the area of the foveal avascular zone and diabetic macular edema. Results. Significant differences in the area of the foveal avascular zone, perimeter of the foveal avascular zone, and perifoveal density in the deep capillary plexus were observed at all three time points. In the fully adjusted linear regression model, those without diabetic macular edema had a reduced probability for changes in the foveal avascular zone at one and three months after surgery (effect estimate *β* = −0.20 [95% CI −0.31; −0.09] and *β* = −0.13 [−0.22; −0.03] for one and three months, respectively) compared with those with diabetic macular edema. Conclusions. Cataract surgery itself does not cause significant and permanent increase in diabetic macular edema three months post-surgery. On the contrary, in a group with diabetic macular edema before the surgery, central retinal thickness tended to stabilize three months after surgery. If the duration of diabetes is shorter and diabetes is better compensated, the possibility of changes in the foveal avascular zone is reduced.

## 1. Introduction

Currently, diabetes and cataracts co-occur frequently, and most of these patients need cataract surgery to restore their eyesight. It is still a question among ophthalmologists whether to treat patients with diabetic macular edema before cataract surgery or whether it can be postponed. Additionally, discussion about complications affecting the macula after cataract surgery is still ongoing.

Diabetes mellitus is the most common risk factor for cataract formation, with a 2- to 5-fold increased risk among diabetes mellitus patients [1]. In patients with type 2 diabetes mellitus, the 10-year incidence of cataract surgery was 24.9% [2]. Although the presence of diabetic retinopathy, which is an indicator of poor ocular condition in diabetes mellitus patients, may influence the incidence of cataracts, this association has been rarely reported [3]. The pathophysiology of cataract development in patients with diabetes is multifactorial; mechanisms include increased accumulation of advanced glycation end-products, increased oxidative stress, and activation of the polyol pathway [4]. With the increasing incidence of cataracts in diabetic patients, cataract surgery is a common procedure that allows sufficient visual rehabilitation. Preoperative evaluation should begin with a standard-of-care cataract evaluation, including an assessment of whether vision loss is attributable to cataracts or diabetic retinopathy, which must be differentiated by a thorough examination as well as other ocular examinations. In addition, systemic blood glucose must be evaluated, and high blood glucose must be treated before cataract surgery to decrease complications during and after surgery [5]. According to Woo et al.’s (2016) survey of ophthalmologists and anesthesiologists, most ophthalmologists recommend cancelling surgery if the preoperative blood glucose level is higher than 410 mg/dL and the glycated hemoglobin (HbA1C) level is higher than 15.9% [6]. The reasonable HbA1c goal for older adults who are otherwise healthy, with few coexisting chronic illnesses and intact cognitive function and functional status, has been determined to be 7.0–7.5% (53–58 mmol/mol) [7].

Cataract surgery per se might exacerbate diabetic retinopathy as a result of blood–aqueous barrier impairment in diabetic patients or cause cystoid macular edema [8]. Cystoid macular edema, also called diabetic macular edema (DME), remains an important cause of suboptimal visual acuity after cataract surgery. Cystoid macular edema is characterized by the accumulation of fluid in the macula due to leakage from capillaries [9]. However, the cause of edema after cataract surgery has not been conclusively determined: “Surgical trauma with an inflammatory response, removal of the crystalline lens, breakdown of the blood-retina barrier, and anterior-posterior and tangential forces of the posterior vitreous membrane are likely causes” [10]. Preoperatively treating DME to achieve stabilization with anti-vascular endothelial growth factor (anti-VEGF) intravitreal injections and laser photocoagulation may improve postoperative outcomes. Additionally, treating eyes with anti-VEGF factor intravitreal injections during or immediately after surgery has been shown to mitigate the risk of DME progression in patients with diabetes [11]. Although patients with DME can undergo cataract surgery, ophthalmologists try to avoid surgery as much as possible to lower the risk of complications. In recent studies among otherwise healthy individuals, changes that occur in the retina due to the surgery are transient. CRT tends to increase in the one-month period after the surgery, with further decrease compared to baseline in the following two- to six-month period [12,13]. According to these results, cataract surgery might not be postponed. However, the correct procedures for cases of DME in diabetes patients undergoing cataract surgery are still unclear.

Currently, one of the methods to evaluate retinal microvasculature is optical coherence tomography angiography (OCTA), which is a noninvasive technological method for the evaluation of retinal blood vessels that is currently used worldwide. OCTA can image the retinal vasculature in eyes with diabetic retinopathy and can detect subclinical microvasculopathy before the onset of clinically observable retinopathy [14]. It has major benefits compared to the gold standard of fluorescent angiography, which has been used for more than 50 years. To perform fluorescein angiography, an intravenous hydrocarbon dye, fluorescein sodium, is injected, but this can lead to life-threatening side effects due to allergic reactions to contrast dye [1]. OCTA allows evaluation of the deep capillary plexus, which is not possible using fluorescent angiography [15]. OCTA has been used to detect early microvascular changes in diabetes mellitus patients, even before clinical signs appear [3]. Microvascular changes on OCTA reflect the patient’s systemic status, as they are significantly correlated with HbA1c levels [16].

Diabetic patients undergoing phacoemulsification surgery exhibited a transient mild reduction in vessel density in the deep capillary plexus one month after surgery, which was correlated with postoperative macular thickening. Macular thickness, as well as vascular density in the superficial capillary plexus, were increased in diabetic patients at one and three months after cataract surgery. Macular thickness may be predictive of postoperative diabetic retinopathy progression [3]. In addition, glycemic control and increased surgical trauma were found to be associated with the degree of deep capillary plexus vascular density reduction [4]. Therefore, investigation of processes related to cataract surgery in diabetic patients is of great importance.

In the past two years, some publications comparing changes in the retina after cataract surgery between diabetic and nondiabetic patients have been published [4,17,18]. However, studies exclusively comparing type 2 diabetes mellitus patients with different macular conditions before surgery are scarce. This study aimed to evaluate changes in the central retina among type 2 diabetes patients undergoing cataract surgery. Groups were formed according to macular condition before surgery. Variables such as the duration of diabetes and diabetes compensation were taken into account. We used OCTA technology in this study.

We posed three main questions:To what degree does uncomplicated cataract surgery affect the retina?Does the severity of diabetic retinopathy and DME affect the postsurgical result?Are preoperative macular conditions associated with the duration of diabetes mellitus, diabetes compensation, and/or the use of insulin?

## 2. Materials and Methods

Patients with type 2 diabetes mellitus were included in this prospective longitudinal study. All patients had planned to undergo small-incision phacoemulsification cataract surgery with intraocular lens implantation at Pauls Stradins Clinical University Hospital in Latvia from October 2020 until August 2021. The study was approved by the Riga Stradins University Committee of Ethics on 13 August 2020 and followed the tenets of the Declaration of Helsinki. All patients provided written informed consent before undergoing surgery. The inclusion criteria for patients who underwent cataract surgery were at least 18 years of age and serologically proven type 2 diabetes mellitus treated by medication (insulin injections or peroral medication). The exclusion criteria were a history of ocular trauma or the presence of another ocular disease that affected the macula (e.g., age-related macular degeneration, glaucoma, severe cataract, or previous vitrectomy). Of the enrolled patients, ten had DME (fluid accumulation in the macular zone, including center- and non-center involving DME), and twenty-four did not have visible DME on optical coherence tomography. This includes DME patients that had clinical (equal to or greater than 290 µm) and subclinical (macular thickness between 260 µm and 290 µm) macular edema [14].

All patients underwent complete ophthalmic examinations, which included best-corrected visual acuity (BCVA) measurement, intraocular pressure measurement, slit-lamp, fundus examination, optical coherence tomography, and optical coherence tomography angiography (OCTA), to evaluate the macula at baseline (the day before cataract surgery) and at one and three months after surgery. Additionally, for those with any changes in the retina, fundus photography was performed. The HbA1c (%) level and duration of type 2 diabetes mellitus were recorded. Depending on visible structural changes in the retina during fundoscopy, diabetic retinopathy was classified as no diabetic retinopathy (no-DR) when no characteristic diabetes-related changes were visible in the fundus and nonproliferative diabetic retinopathy (NPDR) when characteristic changes, such as microaneurysms, dot-blot hemorrhages, cotton wool spots, venous beading, and vascular loops, were observed in the fundus. The most severe form was considered proliferative diabetic retinopathy (PDR), defined as the presence of the previously mentioned changes accompanied by new blood vessels in the fundus and anterior parts of the eye, for example, the anterior chamber angle and iris.

All patients underwent phacoemulsification cataract surgery with intraocular lens implantation. All surgeries were performed by various surgeons, and no intraoperative complications occurred. All patients received dexamethasone, neomycin sulphate, and polymyxin sulphate plus eye drops for 4 weeks after surgery. Patients who previously received intravitreal injections or laser photocoagulation completed their last treatment at least three months before the surgery.

All parameters for microvascular evaluation were measured using an OCTA device (Optovue RTVue XR 100 Avanti Edition. Software Version 2015.0, Optovue, Inc., USA). Macular 3 × 3 mm OCTA images to observe the foveal avascular zone (FAZ) and 6 × 6 mm images to observe vascular density were obtained. Central retinal thickness was measured from the inner limiting membrane to the retinal pigment epithelium. Mean para- and perifoveal vessel densities were measured in the superficial capillary plexus and deep capillary plexus. Additionally, we measured the FAZ and its perimeter. All parameters were measured automatically and only 7/10 quality of scans were included for further research.

Comparisons of basic parameters between groups were performed using conventional nonparametric tests. The Friedman test was used to compare the results in each group separately at different time points, and values are expressed as the median and interquartile range (IQR). The Mann-Whitney test was used to compare the results between groups before and at one and three months after surgery. Multiple linear regression models adjusted for HbA1c level and duration of diabetes mellitus were constructed to investigate the association between the FAZ area and DME. The adjustment set was determined by the univariate relationships between independent factors and the FAZ area. The results are presented as the effect estimate (β) and 95% confidence interval (CI). All the data were analyzed using Statistical Package for Social Sciences (SPSS) 26th version. A *p* value ≤ 0.05 was considered statistically significant.

## 3. Results

Altogether, 32 eyes of 32 type 2 diabetes mellitus patients were analyzed. Groups were established according to macular condition prior to surgery (Table 1).

In patients with DME, the microvascular parameters of FAZ area, FAZ perimeter, BCVA, and central retinal thickness differed significantly among the three time points of the study: at baseline, one month post-surgery, and three months post-surgery. 

In patients without DME, the parameters that significantly differed among the time points were BCVA, FAZ area, FAZ perimeter, mean perifoveal density in the deep capillary plexus (shown in Figure 1, Figure 2 and Figure 3), and central retinal thickness (Table 2).

The characteristics of the groups significantly differed at each time point. Significant results were obtained for BCVA, FAZ area, FAZ perimeter, mean parafoveal vascular density in the deep capillary plexus, and central retinal thickness (Table 3).

In the fully adjusted linear regression models adjusted for HbA1c and duration of diabetes mellitus, those without DME had a reduced probability of changes in the FAZ at the second and third time points; however, at the third time point, this difference was slightly smaller (Table 4).

## 4. Discussion

To understand how uncomplicated cataract surgery affects the retina in type 2 diabetes mellitus patients and if these changes affect post-surgical results after time, various structures were studied during three-month period. As the question is whether to treat DME and postpone cataract surgery if DME is present, we studied changes in various retinal structures and compared them between patients who had DME before the surgery and who did not. Significant results were obtained in the FAZ and deep capillary plexus.

The FAZ is a specialized region of the human retina that usually has the highest cone photoreceptor density and oxygen consumption. In the study of Chandrakumar Balaratnasingam et al., it is suggested that the FAZ has great utility as the structural unit of visual function among patients with diabetic retinopathy [19]. Boned Murillo et al.’s systematic review of aggregated data concluded that in patients with diabetes mellitus (both type 1 and type 2 diabetes mellitus patients) without clinical diabetic retinopathy, the FAZ and areas of capillary nonperfusion were increased; these results suggested that FAZ metrics may have prognostic value in diabetic retinopathy progression, impending DME, and overall visual acuity [20]. In that study, none of the FAZ metrics differed in terms of severity of diabetic retinopathy, indicating that they do not play a crucial role in advanced diabetic retinopathy, though they may have prognostic value in predicting changes in such parameters [20]. Comparing patients with a normal retina to those with macular edema, we found a significant difference in the FAZ area at one and three months after surgery between the two groups. The FAZ area was almost twice as wide in patients with macular edema than in those without edema at one month after surgery, which implies that ischemia was induced by surgery. Although at three months after surgery, the FAZ area in the group with DME had diminished slightly, it was still significantly wider than that in those without DME (Figure 1). The FAZ area decrease cannot be unequivocally attributed to diabetes, because Zhao et al. showed a decrease in the FAZ area that remained three months postoperatively in nondiabetic patients. This may well be because of intraocular pressure fluctuations during surgery, postoperative inflammation, which is a normal reaction after this intervention, and an increase in light exposure during and after cataract surgery [21]. Very similar results were obtained for the FAZ perimeter, although the perimeter was significantly different between the groups before surgery; greater irregularity was detected among patients with DME (Figure 2). In some studies, irregularity of the FAZ perimeter is mentioned as an ischemic factor. In Gildea et al.’s systemic review, the full-thickness FAZ area in type 2 diabetes patients with no retinopathy increased from 0.33–0.41 mm^2^ as diabetic retinopathy deteriorated [22]. However, in our study, the FAZ area in diabetes patients without DME was lower. Visual changes are shown in Figure 4. In diabetic retinopathy patients with DME, the full-thickness FAZ area ranged from 0.47–0.76 mm^2^, which was comparable to that in our study. However, in Feng et al.’s 2021 study, the FAZ area in the diabetic group was smaller than that in the nondiabetic group, but the difference was nonsignificant [3]. Additionally, the FAZ perimeter was very similar in both groups. This suggests that this metric is controversial.

A significant difference in the mean perifoveal vascular density in the deep capillary plexus at baseline and at three months after surgery was found, with diminished vascular density in patients with DME (Figure 3). In Feng et al.’s study, the vascular density at baseline in the control group was very similar to that in the diabetic group. Additionally, in our study, the results in patients without DME three months after surgery were very similar to those in the diabetes group three months after surgery [3]. Wang et al. noted a transient vascular density reduction in the deep capillary plexus thirty days post-surgery that returned to normal ninety days later. In patients with diabetes, no significant change in the superficial capillary plexus vascular density ninety days post-surgery was noted [4]. In our study, at one month after surgery, no significant difference was found between groups. In both groups, vascular density increased over time, which confirms results from other studies that showed that vascular density tended to restore over time. Compared with the results of Feng et al.’s study, our results showed that vascular density continued to increase, even at three months after surgery as shown in Table 2. In their study, it tended to be increased at only one and seven days after surgery and had decreased at one month after surgery [3]. No significant differences in changes in the para- and perifoveal superficial capillary plexus areas were found; this is in contrast with their study that found a difference in the superficial capillary plexus but not the deep capillary plexus, and their follow-up period was also three months [3]. Our results also differed from those of Hui Li et al.’s study, in which a significant increase was observed in the deep capillary plexus and superficial capillary plexus. The authors considered that inflammation impacted the assessment of density parameters [23]. It is possible that the vascular density increase is due to an inflammatory response causing vascular permeability and the presence of inflammatory factors that are increased in the eyes of patients with diabetes mellitus. Interestingly, perifoveal vessel density in the DCP in non-DME groups showed significant increase as shown in Figure 3, but there was no difference in the DME group, which could be explained that in the non-DME group blood vessels are more sensitive to changes during surgery, while persistent edema around the vessels might suppress vessel sensitivity. Additionally, abnormalities in retinal hemodynamics play a role in microcirculation disturbances, which increased compensational mechanisms of superficial vasculature [3]. Hui Li et al. showed that changes in the retinal microvasculature can be present in the absence of diabetic retinopathy. The vessel density of the deep retinal vascular layer was negatively correlated with fasting blood glucose; these indicators are helpful for endocrinologists and ophthalmologists detecting early retinal lesions in diabetes patients [24].

A significant difference in the central retinal thickness at one month after surgery was observed; central retinal thickness was significantly higher in DME patients, which indicated that DME tended to increase for one month after surgery, considering that surgery itself increases the exacerbation of DME. Although central retinal thickness tended to increase in both groups, at three months after surgery, it diminished in the control group; this indicated that the degree of DME was less in the exposure group after surgery than it was before surgery and that it was slightly greater in the control group, considering that cataract surgery does not worsen diabetic retinopathy and the risk of DME in type 2 diabetes mellitus patients, as described in Le Feng et al. [3]. Overall, at three months after surgery, DME diminished; therefore, we can conclude that cataract surgery in patients with or without DME does not adversely affect outcomes. As most of the previous publications suggest, DME and diabetic retinopathy should be treated before scheduling cataract surgery. DME, a predictor of poor visual outcomes, should be treated with laser photocoagulation or intravitreal injections of either anti-VEGF agents or corticosteroids [11,24,25]. Although these recommendations have existed for some time and ophthalmologists are aware, most of the patients who had fluid accumulation in the macula before surgery did not have enough money to pay for anti-VEGF injections. It is interesting to contemplate the potential results if we had evaluated patients for more than three months—would there be further improvement or not?

Diabetes mellitus severity is often estimated by HbA1c, which reflects the percent of erythrocytes that have undergone the nonenzymatic linkage of glucose to their proteins over the lifetime of the cell and is considered one of the significant indicators of the course of diabetes [26]. Most of the patients included in the study did not know that they ought to monitor their diabetes compensation every three months, as suggested by the European Endocrinology Guidelines. Only about half the patients knew what level they were aiming to achieve; however, they were still asked to enroll in the study. This is likely the explanation for why the study group had poorly compensated diabetes. Even the controls had uncompensated diabetes, although the average value was closer to the recommended value.

Mostly, our results correspond to similar research. The results tend to move towards the fact that cataract surgery should be conducted to renew eyesight whether DME is or is not present at the time. Treatment for DME can be carried out before and continued after the surgery. The research should be continued for longer period of time to evaluate whether vascular density returns to baseline values. More patients should be included in further research, especially in a group with DME before the surgery to answer the main question whether changes in the retina caused by cataract surgery are permanent or vanish with time. Whether the macula eventually returns to baseline values after cataract surgery remains to be discovered.

## 5. Conclusions

Uncomplicated cataract surgery does affect the retina, but most changes are reversible with time. The FAZ area significantly increased in the group with DME before the surgery, but tended to decrease three months after, implicating that ischemic changes are induced by surgery itself. Our findings show that perifoveal vascular density in the deep capillary plexus tended to increase at one and three months after cataract surgery, despite the presence of DME before surgery or not, suggesting that changes in vascular density were due to the surgery itself and tend to normalize within three months post-operation. The deep capillary plexus tended to be more sensitive to hemodynamic and inflammatory changes after cataract surgery. Cataract surgery itself does not cause significant and permanent increase in diabetic macular edema three months post-surgery. On the contrary, if DME was present before the surgery, CRT tended to stabilize three months after surgery. Macular edema does not prohibit surgery because surgery does not cause long-term DME. In addition, a shorter duration of diabetes and better diabetes compensation leads to a lower possibility of changes in the FAZ area, which is one of the indicators for ischemia in the central retina area, at one and three months after surgery.

## Figures and Tables

**Figure 1 diagnostics-13-00605-f001:**
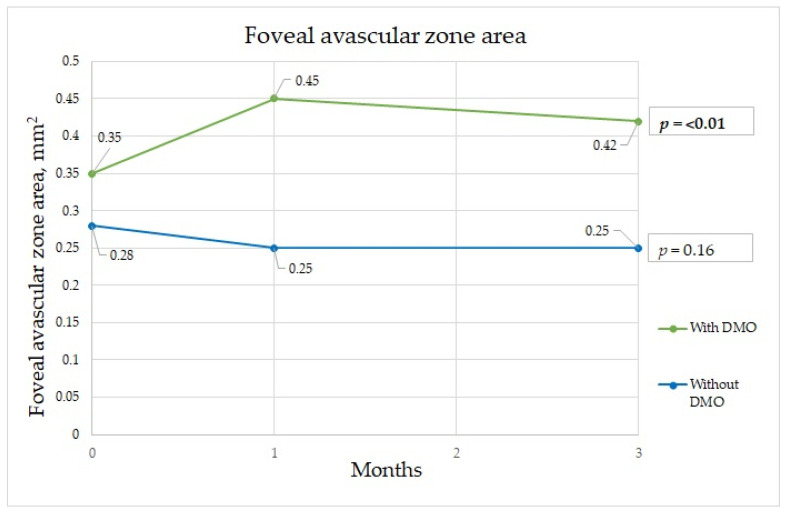
Foveal avascular zone changes during three-month period.

**Figure 2 diagnostics-13-00605-f002:**
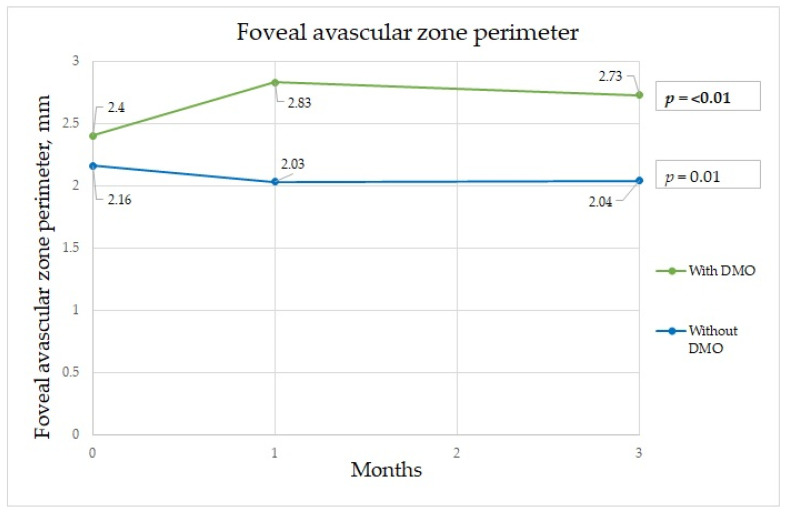
Foveal avascular zone perimeter changes during three-month period.

**Figure 3 diagnostics-13-00605-f003:**
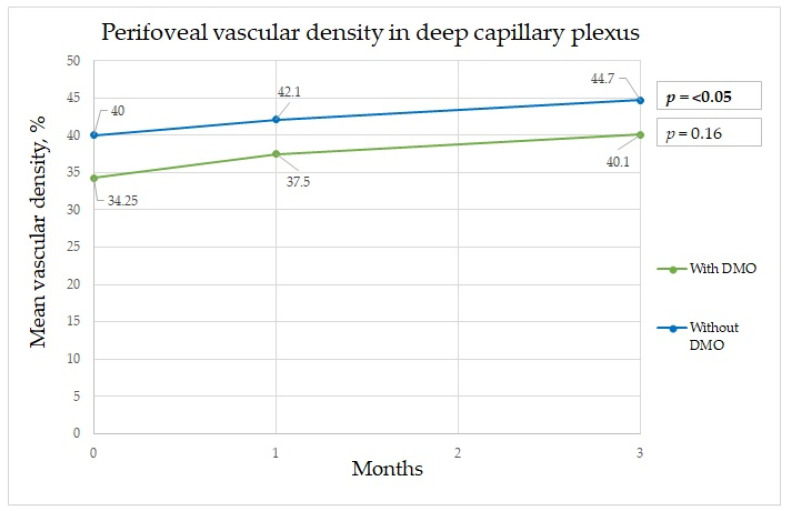
Perifoveal vascular density changes in deep capillary plexus during three-month period.

**Figure 4 diagnostics-13-00605-f004:**
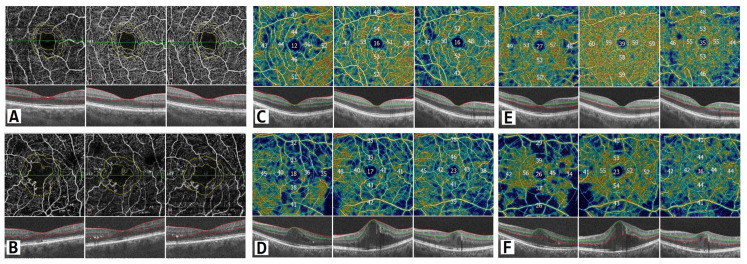
Vascular changes on OCT angiography. The scan area is 6 mm × 6 mm. Upper panel: patients without DME; lower panel: patients with DME. (**A**,**B**) FAZ changes showing distorted circularity among DME patients. (**C**,**D**) Vascular density changes in the superficial capillary plexus in parafoveal and perifoveal regions. (**E**,**F**) Vascular density changes in the deep capillary plexus in parafoveal and perifoveal regions during the three-month period. In color-coded maps warmer colors represent higher vascular density and numbers show vascular density in percentage.

**Table 1 diagnostics-13-00605-t001:** Baseline characteristics of the study participants.

Variable	With DME (10 (31.2%))	Without DME (22 (68.8%))	*p* Value
Sex (male)	1 (16.7%)	5 (83.3%)	0.42
Type of diabetic retinopathy			
No-DR	0 (0.0%)	15 (100%)	<0.01
NPDR	9 (56.3%)	7 (43.8%)	
PDR	1 (100%)	0 (0.0%)	
Age (years), median (IQR)	68.00 (63.25–74.50)	71.00 (64.00–77.00)	0.32
HbA1C, %, median (IQR)	8.68 (7.89–9.20)	7.00 (6.39–8.79)	0.06
Duration of diabetes mellitus (years), median (IQR)	15.50 (9.50–22.25)	11.50 7.25–20.75)	0.42
Use of insulin (years), median (IQR)	0.00 (0.00–6.50)	0.00 (0.00–6.50)	0.72

DME, diabetic macular edema; DR, diabetic retinopathy; NPDR, nonproliferative diabetic retinopathy; PDR, proliferative diabetic retinopathy; HbA1C, glycated hemoglobin.

**Table 2 diagnostics-13-00605-t002:** Comparisons of parameters at baseline, one month post-surgery, and three months post-surgery in patients with and without DME at baseline.

		Baseline, Median (IQR)	One Month Post-Surgery, Median (IQR)	Three Months Post-Surgery, Median (IQR)	*p* Value
Patients with DME (*n* = 10)	BCVA (logMAR)	0.6 (0.15–0.3)	0.35 (0.72–0.19)	0.2 (0.64–0.15)	0.02
	FAZ area (mm^2^)	0.35 (0.30–0.40)	0.45 (0.39–0.55)	0.42 (0.29–0.50)	<0.01
	FAZ perimeter (mm)	2.40 (2.29–2.30)	2.83 (2.61–3.50)	2.73 (2.35–3.00)	<0.01
	Mean parafoveal density in SCP (%)	38. 5 (36.3–43.85)	39.6 (32.42–43.02)	43.55 (35.33–49.08)	0.42
	Mean parafoveal density in DCP (%)	43.60 (36.28–47.38)	42.35 (40.43–50.65)	43.70 (39.75–47.50)	0.87
	Mean perifoveal density in SCP (%)	38.60 (36.60–40.45)	39.15 (36.93–45.35)	42.65 (38.55–47.58)	0.07
	Mean perifoveal density in DCP (%)	34.25 (32.30–39.00)	37.50 (36.40–43.25)	40.10 (36.60–46.25)	0.16
	Central retinal thickness (µm)	290.50 (225.50–361.50)	300.50 (266.25–452.25)	286.50 (242.00–323.75)	0.05
Patients without DME (*n* = 22)	BCVA (logMAR)	0.52 (0.70–0.38)	0.10 (0.25–0.05)	0.1 (0.16–0.00)	<0.01
	FAZ area (mm^2^)	0.28 (0.23–0.36)	0.25 (0.18–0.34)	0.25 (0.20–0.34)	0.16
	FAZ perimeter (mm)	2.16 (1.94–2.47)	2.03 (1.79–2.36)	2.04 (1.85–2.39)	0.01
	Mean parafoveal density in SCP (%)	42.00 (39.60–45.20)	44.45 (38.02–47.88)	43.30 (40.60–47.60)	0.91
	Mean parafoveal density in DCP (%)	48.90 (41.00–51.90)	48.70 (45.63–54.20)	51.00 (47.73–54.23)	0.14
	Mean perifoveal density in SCP (%)	42.70 (38.15–57.65)	42.60 (39.30–47.20)	43.70 (41.80–47.20)	0.99
	Mean perifoveal density in DCP (%)	40.00 (34.28–46.35)	42.10 (37.80–47.95)	44.70 (37.20–49.00)	<0.01
	Central retinal thickness (µm)	256.00 (249.00–275.00)	269.00 (251.50–283.25)	268.50 (256.50–283.75)	<0.01

BCVA, best-corrected visual acuity; SCP, superficial capillary plexus; DCP, deep capillary plexus; FAZ, foveal avascular zone; DME, diabetic macular edema; M, median, IQR, interquartile range.

**Table 3 diagnostics-13-00605-t003:** Comparisons of parameters at three time points: at baseline and at one and three months post-surgery.

	Baseline	One Month Post-Surgery	Three Months Post-Surgery
Best-corrected visual acuity	0.56	0.018	<0.01
FAZ area (mm^2^)	0.16	<0.01	<0.01
FAZ perimeter (mm)	0.034	<0.01	<0.01
Mean parafoveal density in SCP (%)	0.14	0.11	0.94
Mean parafoveal density in DCP (%)	0.025	0.07	<0.01
Mean perifoveal density in SCP (%)	0.06	0.19	0.66
Mean perifoveal density in DCP (%)	0.10	0.19	0.26
Central retinal thickness (µm)	0.11	0.04	0.44

FAZ, foveal avascular zone; SCP, superficial capillary plexus; DCP, deep capillary plexus.

**Table 4 diagnostics-13-00605-t004:** Association between DME and the FAZ area in both groups at one and three months post-surgery.

Time Points	Variable	Effect Estimate, *β*	95% Confidence Interval	*p* Value
One month post-surgery	Patients with DME vs. patients without DME	−0.20	−0.31; −0.09	<0.01
Three months post-surgery	Patients with DME vs. patients without DME	−0.13	−0.22; −0.03	0.01

## Data Availability

The data used to support the findings of this study are available from the corresponding author upon reasonable request.

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
