# Peer review of "Microstructural Changes in the Macula Following Cataract Surgery in Patients with Type 2 Diabetes Mellitus Detected Using Optical Coherence Tomography Angiography"

_diagnostics, 2023, doi:10.3390/diagnostics13040605_

Round 1

Reviewer 1 Report

Title: “Microstructural changes in the macula following cataract surgery in patients with type 2 diabetes mellitus detected using optical coherence tomography angiography.”

In this work thirty-four type 2 diabetes mellitus patients who underwent phacoemulsification cataract surgery were included in a prospective longitudinal study. Among them, 29.4% had macular edema before surgery, while 70.6% had a normal macular structure. All these patients underwent ophthalmic examinations, including optical coherence tomography angiography, at baseline and at one and three months after surgery. A Mann‒Whitney test was performed to compare the area of the foveal avascular zone, the perimeter of the foveal avascular zone, as well as the mean vascular density in the para- and perifoveal deep and superficial capillary plexuses. In particular, all parameters were measured before and at one and three months after surgery. Multiple linear regression models with adjustments for glycated hemoglobin and duration of diabetes mellitus were constructed to assess the association between the area of the foveal avascular zone and diabetic macular edema. Through these procedures a significant difference in the area of the foveal avascular zone, perimeter of the foveal avascular zone and perifoveal density in the deep capillary plexus were observed at all three time points. More specifically, within the fully adjusted linear regression model, the patients without diabetic macular edema had a reduced probability of changes in the foveal avascular zone at one and three months after surgery (effect estimate β = -0.20 [95% CI -0.31; -0.09], and β = -0.13, [-0.22; -0.03], for one and three months, respectively) with respect to those with diabetic macular edema. The authors aim to conclude that cataract surgery itself does not cause significant and permanent increase in diabetic macular edema three months post-surgery. However, in a group with diabetic macular edema before the surgery, central retinal thickness tended to stabilize three months after surgery.

General comment: Although this work seem to be interesting it should be revised to further increase its quality and impact. In particular, the authors should present their results in a better way (i.e., not using only tables). Can they transform them in plots ? In this way interested readers could understand in a better and faster way the main achievements of this work. In general, figures should be provided in high resolution and resized (enlarged). Figure captions should be rewritten and improved to be more informative.

Some detailed comments:

Figure 1. Vascular changes on OCT angiography. A, B: FAZ changes; C, D: vascular density changes in the superficial capillary plexus; E, F: vascular density changes in the deep capillary plexus during the three-month period. Upper panel: patients without DME; lower panel: patients with DME.

*) This figure should be enlarged and, eventually, labels should be inserted into. The figure caption should be improved and rewritten to be more informative.

Table 2. Comparisons of parameters at baseline, one month post-surgery, and three months post- surgery in patients with and without DME at baseline.

*) This table seems to be crucial for the work. However, it seems that to provide all this information though a table is a suboptimal solution. The authors should provide some plots underlining the main achievements. Please rework.

4. Discussion

5. Conclusions

*) These sections should be improved to better explain what are the main achievements of this work and what are the questions this work aim to answer and the still open questions.

Reviewer 2 Report

The number of included patients is quite limited (10 and 22), as diabetes and cataract are not a rare diseases.

References are not properly presented according to the journal requirements and are not completed, only first name of authors  et al. is written. Also not only articles in journals but web pages are included.

For example:

·        Balasopoulou et al., “Symposium Recent advances and challenges in the management of retinoblastoma Globe - saving Treat- 390
ments,” BMC Ophthalmol., vol. 17, no. 1, p. 1, 2017, doi: 10.4103/ijo.IJO.

·        A. Wyle and E. Wyle, “Optical Coherence Angiography by Tsukuba.pdf.”

·        M. Grau et al., “Optical Coherence Tomography Angiography in Diabetic Patients: A Systematic Review,” 2021, doi: 10.3390/bi- 401omedicines10010088.

·        D. Care and S. S. Suppl, “Summary of Revisions : Standards of Medical Care in Diabetes d 2021,” vol. 44, no. January, pp. 4–6, 376
2021.

The names of first authors of references in the text do not match names in the reference list.

There are 28 references in the list, but in the text the last number is 24.

In the discussion (line 224)  it is written „FAZ has great utility as a biomarker of visual function among patients with diabetic retinopathy”, FAZ is not biomarker of the visual function, rather the structure of the retina.

Round 2

Reviewer 2 Report

Changes have been made according to comments.